# RELATIVE SCALING LAWS FOR LLMS

## ABSTRACT

Scaling laws describe how language models improve with additional data, parameters, and compute. While widely used, they are typically measured on aggregate test sets. Aggregate evaluations yield clean trends but average over heterogeneous subpopulations, obscuring performance disparities. We introduce **relative scaling laws**, which track how performance gaps between test distributions evolve with scale rather than focusing solely on absolute error. Using 255 decoder-only Transformers trained under matched-compute (*IsoFLOP*) budgets from $10^{18}$–$10^{20}$ FLOPs on standard pretraining datasets, we find diverse trajectories: academic domains on MMLU converge toward parity; regional English dialects shift depending on population size; and clusters of AI risk behaviours split, with capability- and influence-related risks increasing during pretraining while adversarial risks do not. These results show that although scaling improves overall performance, it is not a universal equalizer. To support further study, we release all model checkpoints from this work to enable practitioners to measure *relative* alongside traditional scaling laws, in order to better prioritize robustness challenges in light of the bitter lesson[1].

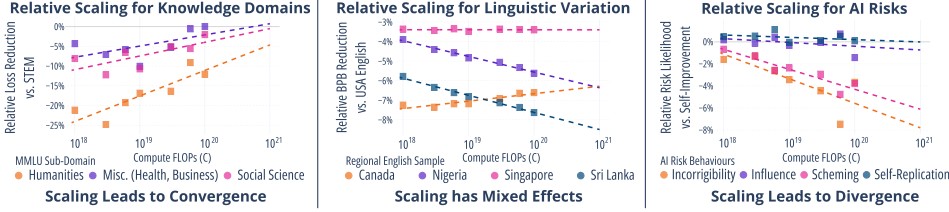

Figure 1: **Relative scaling law case studies.** Scaling compute has uneven effects (illustrated here with models trained on DCLM (Li et al., 2024) from $10^{18}$–$10^{20}$ FLOPs): (left) knowledge domains, (center) English variation, and (right) AI risk behaviours. We propose relative scaling laws as a method to measure which gaps close with scale and which persist or widen.

## 1 INTRODUCTION

Neural scaling laws show that language model error typically decreases as a power law with increases in model size, data, and compute (Hestness et al., 2017; Kaplan et al., 2020; Hoffmann et al., 2022). These trends suggest that "bigger is better", with only rare cases of inverse scaling (McKenzie et al., 2023; Sharma et al., 2024). However, because these laws average over heterogeneous test distributions, the *rate* of improvement may not be uniform across subdomains (Magnusson et al., 2024). In practice, gains from scale may favor some areas more than others, much as economic growth can deliver uneven returns across groups and increase inequality (Piketty, 2015).

We introduce *relative scaling laws* to study this dimension of scaling effects. Whereas traditional scaling laws describe absolute improvements, relative scaling laws quantify how *performance gaps* between settings evolve with scale. This separates disparities at small scales – often shaped by confounding factors such as inherent data entropy — from differences in improvement rate, which more directly capture the response to scale. The relative law is fit directly as a power law by regressing the ratio of treatment to baseline error on compute. This procedure is no harder than fitting absolute laws, but indicates whether gaps persist, narrow, or widen as compute increases. This provides

---

[1] All models to be released on HuggingFace upon publication to abide by anonymity constraints and file size constraints in supplementary material.

a concrete lens on distributional consequences of scaling model compute , with implications for robustness, fairness, and risk.

To support such analyses, we train 255 decoder-only Transformers under matched-compute (*IsoFLOP*) budgets from $10^{18}$ to $10^{20}$ FLOPs, consisting of 85 models on each of three pretraining datasets. Training under fixed compute ensures that comparisons reflect the tradeoff between model size and data size, avoiding confounds that otherwise complicate scaling-law studies (Hoffmann et al., 2022; Besiroglu et al., 2024). The datasets span three distinct design philosophies— permissively licensed corpora, filtered web data, and hybrid web+synthetic mixtures—so that we can test whether scaling trends generalize across training data sources. We release the full model suite, providing a resource analogous to Biderman et al. (2023) for downstream scaling-law evaluation (Roberts et al., 2025; Hu et al., 2024).

Finally, we demonstrate the scope of relative scaling laws in three case studies. First, we analyze MMLU (Hendrycks et al., 2021) sub-domains to measure how knowledge scales across academic disciplines. Second, we evaluate robustness to English variation, testing generalization across regional English using the International Corpus of English (ICE) (Greenbaum, 1996). Third, we assess how relative risks emerge during pretraining using Anthropic's AI risk evaluations from Perez et al. (2023). Across all these settings, we fit both traditional and relative scaling laws.

**Contributions.** Our contributions combine conceptual, resource, and empirical components:

1. **Relative scaling framework.** We formalize *relative scaling laws*, which separate initial disparities from differences in improvement rate. Formulated as a power law, relative scaling provides a clear diagnostic of which distributions benefit the most from scaling.

2. **Open-source scaling suite.** We train and release 255 decoder-only Transformers under IsoFLOP budgets from $10^{18}$–$10^{20}$ FLOPs across three corpora—COMMONPILE (Kandpal et al., 2025), DCLM BASELINE (Li et al., 2024), and NEMOTRON-CC (Su et al., 2025). The suite enables reproducible study of both traditional and relative scaling laws.

3. **Empirical case studies.** We apply relative scaling laws to three domains: academic knowledge (Massively Multitask Language Understanding benchmark; MMLU), linguistic variation (International Corpus of English; ICE), and AI risk (Anthropic Advanced AI Risk). Together, these studies show a range of relative scaling effects highlighting the non-uniformity of scale's impacts on distributional robustness.

## 2 RELATIVE SCALING LAWS

Relative scaling laws follow directly from the assumptions of classical scaling laws. Absolute error $E$ is assumed to decrease as a power law in scale $F$ (e.g., FLOPs, tokens, or parameters),

$$E(F) = \alpha F^{-\beta},$$

with $\alpha > 0$ as the initial error level and $\beta \geq 0$ as the rate of improvement with scale (Kaplan et al., 2020). These constants are empirically fit based on sample populations of training runs.

In order to relativize performance gains, we compare two conditions: a *baseline* (the reference, here the most favored under current practice) and a *treatment* of interest. Their relative error $G$ is

$$G(F) = \frac{E_{\text{treatment}}(F)}{E_{\text{baseline}}(F)} = \gamma F^{\Delta\beta}$$

where $\gamma = \alpha_{\text{treatment}}/\alpha_{\text{baseline}}$ captures the initial disparity and $\Delta\beta = \beta_{\text{baseline}} - \beta_{\text{treatment}}$ the difference in improvement rates. If $\Delta\beta < 0$, the treatment improves faster and the gap narrows; if $\Delta\beta > 0$, it improves more slowly and the gap widens; if $\Delta\beta = 0$, the gap remains constant[2].

This form parallels the subgroup laws of Rolf et al. (2021), who model subgroup loss as a mixture of power-law terms for in-group and total data. Our formulation is looser — we do not require subgroup allocations — but the sign of $\Delta\beta$ still forecasts whether gaps shrink or persist. While relative loss can correspond to small absolute differences at low loss, small absolute loss gaps can

---

[2]In this work, we only interpret the slope if the sign is significant at $P < 0.05$ by a bootstrap significance test. We recommend this as a best practice for interpreting $\Delta\beta$.

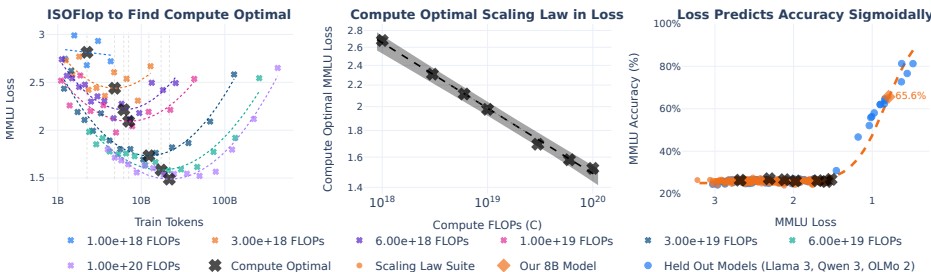

Figure 2: **Compute-optimal scaling and downstream forecasting. Left:** For each FLOP budget, we sweep token and model size to select the compute-optimal token count. **Middle:** Along these compute-optimal points, we estimate how task or subgroup loss scales as a function of compute. **Right:** We show this loss correlates tightly with accuracy sigmoidally, allowing loss to serve as a proxy for downstream progress while measuring effects at reduced scale.

lead to large differences in downstream utility for large scale models (Wei et al., 2022; Du et al., 2024) which motivates this scale-invariant metric rather than absolute disparity (Yeh et al., 2024)[3].

Note that relative scaling laws inherit the assumptions of the absolute form: approximately log-linear behavior, a sufficient range of scales, and consistent evaluation with relatively low variance from factors other than scale. If these assumptions break down, estimates of $\Delta\beta$ may be unstable. Similar to traditional scaling laws, they are therefore best treated as empirical diagnostics, with clear advantages over evaluation at a single scale, rather than fundamental laws.

## 3 RELATIVE SCALING LAW FOUNDATIONS

To study relative scaling, we need a robust foundation for training and evaluation. This section outlines how we construct compute-controlled model families and design evaluation protocols that yield predictable, interpretable scaling curves. Together, these provide the basis on which both traditional and relative scaling laws can be fit and trusted to forecast downstream performance.

### 3.1 ISOFLOP MODEL TRAINING

We train models using the Qwen 3 architecture (Yang et al., 2025) under fixed compute (IsoFLOP) budgets ranging from $10^{18}$ to $10^{20}$ FLOPs. While IsoFLOPs are not strictly necessary for scaling laws, prior work (DeepSeek-AI et al., 2024; Grattafiori et al., 2024; Roberts et al., 2025) has argued that the IsoFLOP-based approach from Hoffmann et al. (2022), shown on the left in Figure 2, is more stable and therefore less exposed to reproducibility issues than alternative formulations from Hoffmann et al. (2022) which regress on a larger number of terms at once (Besiroglu et al., 2024).

Scaling models should be trained such that performance variance is primarily explained by compute, model size, and data size. Without consistent hyperparameter tuning, scaling outcomes can be meaningfully confounded (Porian et al., 2025). Since a full grid search is infeasible, we generalize a tuned configuration (Wen et al., 2025) using heuristic reparameterizations.

Our approach follows two principles: (i) hyperparameters should be explicit functions of model width and FLOP budget; and (ii) training should be stable across runs, since instabilities such as loss spikes would introduce noise into scaling comparisons. We cover the full range of reparameterizations for both architectural and optimizer hyperparameters in Appendix A.

**Training Data.** We train models with the same configuration across three datasets to reflect different pretraining data distributions. COMMONPILE (Kandpal et al., 2025) includes only permissively licensed data, downsampling non-permissive web sources in favor of public domain and openly licensed material. In contrast, the DCLM BASELINE (Li et al., 2024) is drawn entirely from web crawl data but filtered and deduplicated to isolate a high-quality subset. Finally, NEMOTRON-CC (Su et al., 2025) combines large-scale real web data with synthetic rephrasings, representing

---

[3]Beyond test-distribution disparities, relative scaling can be used to compare modeling methods; see App. B.

a hybrid of natural and synthetic text. Comparing scaling behavior across these settings enables assessments of the role of training data in relative scaling results.

## 3.2 EVALUATION PROTOCOLS FOR SCALING LAW ANALYSIS

Reliable scaling law evaluation requires careful elicitation design. Pretraining loss typically follows predictable power laws (Kaplan et al., 2020; Hoffmann et al., 2022), but downstream metrics behave inconsistently: some report smooth scaling in aggregate (Gadre et al., 2024), while others find erratic task-specific trends (Lourie et al., 2025). We find these discrepancies arise largely from evaluation choices — especially prompt formats and metric definitions — which introduce thresholding artifacts (Schaeffer et al., 2023) and surface form competition (Holtzman et al., 2022).

To address this, we first run ablations on prompt formats to identify consistent ones that yield smooth scaling laws without diminishing accuracy. Then, following recent recommendations (Grattafiori et al., 2024; Bhagia et al., 2025), we identify protocols that produce predictable loss curves and reliable compute–loss correlations. In this section, we focus on MMLU (Hendrycks et al., 2021), a widely used benchmark claimed to exhibit unpredictable emergence. Contrary to those claims, we find that with suitable protocol, MMLU scales smoothly and loss correlates strongly with accuracy.[4]

### 3.2.1 PROMPT FORMATS

Evaluation of language models typically takes three forms: (i) open-ended generation or token log-probabilities, (ii) multiple-choice question answering, and (iii) binary classification. Raw log-probabilities scale smoothly by default, but hard metrics like accuracy or pass@1 suffer from thresholding effects that obscure predictability (Schaeffer et al., 2023; 2025). Soft metrics such as conditional log-probabilities reduce thresholding but are noisy due to surface form competition (Holtzman et al., 2022). To obtain reliable scaling laws, evaluation must avoid both problems.

For standard language modeling, perplexity and log-likelihood curves remain smooth without intervention (Magnusson et al., 2024). For binary classification, prior work constrains completions to "yes/no" (Ganguli et al., 2023; Perez et al., 2022), eliminating surface form ambiguity.

In multiple-choice tasks like MMLU, however, elicitation formats have not been standardized. Existing frameworks use either MCQ with letter labels or continuation form (CF) (Gu et al., 2025; Biderman et al., 2024), and sometimes both. MCQ scores accuracy but yields poor loss predictability; CF yields smoother loss but lower accuracy. To address this, we adopt a modified format which includes labels and options as in MCQ, but probabilities are computed over the full label+option strings. As shown in Figure 3,

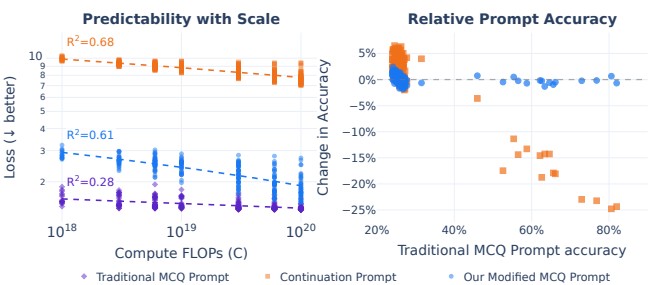

Figure 3: **Prompt formatting drives scaling smoothness. Left:** Degree of variance explained by scale under different prompts. **Right:** Accuracy differences between prompt variants and MCQ.

this method balances predictability and accuracy: CF yields smoother loss curves but lower accuracy ($R^2 = 0.68$, max 57.7%), while MCQ achieves high accuracy but poor loss predictability ($R^2 = 0.28$, max 82.0%). Our modified MCQ format achieves both ($R^2 = 0.61$, max 81.3%), preserving nearly the full accuracy of MCQ while recovering much of the predictability of CF[5].

### 3.2.2 FORECASTING DOWNSTREAM TASK PERFORMANCE USING SCALING LAWS

While loss scales predictably, downstream accuracy often does not (Wei et al., 2022). Similar to prior work measuring downstream capability scaling (Held et al., 2025; Ye et al., 2025; Schaeffer et al., 2025), we therefore utilize loss as our primary metric of interest. However, reliable loss scaling is only meaningful if it forecasts hard metrics such as accuracy.

---

[4]In App. C, we show that the same principles lead to reliable scaling for a variety of other tasks.

[5]All prompt formats are illustrated in detail in App. D

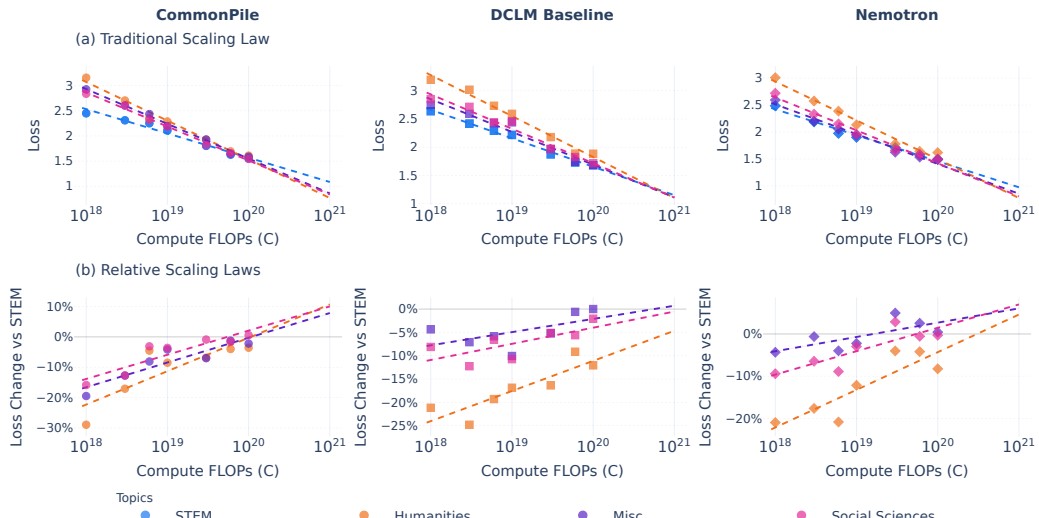

Figure 4: **Relative scaling laws across domains in MMLU.** Columns show results for Common-Pile, DCLM Baseline, and Nemotron. (a) Traditional scaling laws for bits per byte (BPB) scaling across topic groups. (b) Relative scaling laws, normalized so that each curve is expressed relative to the STEM scaling trend. Curves for STEM, humanities, social sciences, and miscellaneous domains converge toward 0 as compute increases, indicating that domain disparities shrink with scale.

To establish this connection, we adopt the *two-stage* procedure of Grattafiori et al. (2024): first fit compute-loss scaling on soft metrics, then map loss to accuracy via a calibration function (typically linear or sigmoid). The first step is a true scaling law, using only compute-optimal runs from our scaling suite. The second can be done observationally (Ruan et al., 2024), allowing us to compare calibration functions against a variety of open-weights models.

Figure 2 shows the effectiveness of this two-step regression using our scaling suite and an internally trained 8B model to fit regressions, with OLMo 2 (OLMo et al., 2025), Llama 3 (Grattafiori et al., 2024), and Qwen 3 (Yang et al., 2025) serving as held-out data. Ultimately, we find that loss can be predicted reliably as a function of compute within our internal models, and accuracy can be predicted reliably as a function of loss across both internal and external models. This establishes accuracy as a predictable function of compute at large scales, while allowing compute–loss scaling to serve as the foundation for downstream scaling-law analysis at smaller scales.

## 4 CASE STUDIES OF RELATIVE SCALING

We demonstrate the scope of relative scaling laws through three case studies: *knowledge domains* (MMLU), where performance converges across disciplines; *regional language variation* (Global Englishes), where some regions converge, others diverge, and some remain unchanged; and *AI risk behaviours*, where certain risks become less likely relative to others as scale increases. Together, these case studies provide evidence of the diverse trajectories that relative scaling laws can reveal.

### 4.1 RELATIVE SCALING FOR KNOWLEDGE DOMAINS

Scaling laws are often interpreted to suggest that sufficiently large models might approach general intelligence, particularly if error decreases across a wide set of tasks, including those not directly emphasized in training. A central question for this perspective is whether all knowledge domains scale equally well, or whether models become increasingly specialized in well-represented topics.

In Figure 4, to examine this question we evaluate scaling laws for MMLU. Panel (a) shows the expected pattern: loss decreases smoothly with compute across STEM, Humanities, Miscellaneous, and Social Sciences for all three training datasets (CommonPile, DCLM Baseline, Nemotron). For example, in the Nemotron run, STEM loss falls from 2.45 at $10^{18}$ FLOPs to 1.56 at $10^{20}$, while Humanities drops from 3.16 to an expected 1.61. Each domain follows the familiar log-linear trend.

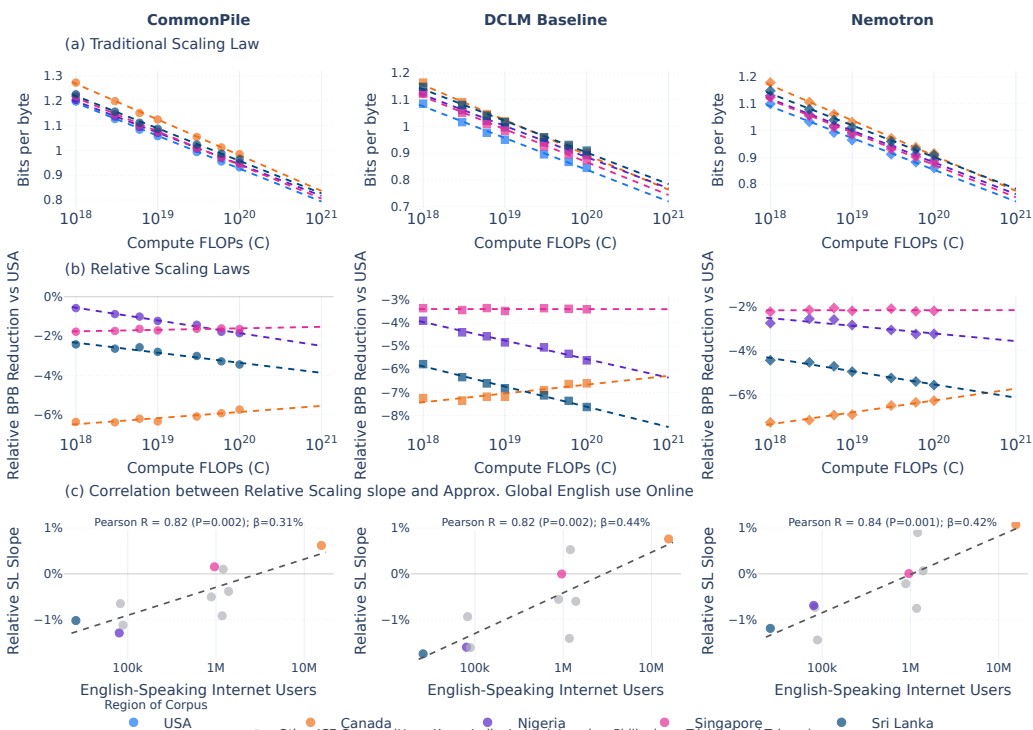

Figure 5: **Relative scaling of written Global Englishes.** Columns show results for CommonPile, DCLM Baseline, and Nemotron. (a) Traditional scaling laws for bits per byte (bpb) vs. compute. (b) Relative scaling laws as bpb differences from U.S. English (dashed line). (c) Correlation between relative scaling slopes and English-speaking internet users at the time the International Corpus of English was collected. Regions with larger online English-speaking populations scale faster.

Panel (b) provides the relative perspective by plotting loss change against STEM. Here we see clear signs of convergence. When trained on the CommonPile, which heavily samples academic work from many disciplines as well as public legal documents, all other subjects have far higher loss than STEM at $10^{18}$ FLOPs ($-29\%$ Humanities, $-16\%$ Social Science, $-19\%$ Misc.), but all converge to within $5\%$ of STEM performance at $10^{20}$ FLOPs.

By comparison, the two web-focused corpora show distinct initial biases towards Misc. (which includes health and business) and away from the Humanities (which includes law and philosophy), but the same trend towards convergence. Under the DCLM Baseline, Humanities, Social Sciences, and Misc. begin $-21\%$, $-8\%$, and $-4\%$ below STEM, narrowing to $-12\%$, $-2\%$, and parity by $10^{20}$ FLOPs; under Nemotron-CC, the gaps shrink from $-21.0\%$, $-9\%$, and $-8\%$ to $-4.0\%$ for Humanities and parity for both others. These trajectories are consistent with expectations based on the domain biases of each corpus: web scrapes only contain sparse sound legal and philosophical material compared to the public data from the Common Pile.

Notably, as models scale, performance imbalances diminish regardless of the underlying training distributions, and all domains converge towards similar performance. These results highlight that while pointwise comparisons at small scales could suggest models are disproportionately STEM-focused, both traditional and relative scaling laws indicate that domain disparities on MMLU are subject to the bitter lesson: with enough compute, the gap narrows naturally.

## 4.2 RELATIVE SCALING FOR LANGUAGE VARIATION

Generalization to new user populations is another key distribution shift. In multilingual settings, performance tracks pretraining representation, with family-level sampling ratios predicting cross-entropy across scales (He et al., 2024). Within-language variation, previously studied in Rae et al. (2022), is subtler due to transfer and interference. We evaluate with the International Corpus of

English (ICE) (Greenbaum, 1996), which includes ∼1M words per variety (500 texts of ∼2,000 words) spanning spoken and written registers under consistent national sampling from speakers with high-school or higher levels of education. We restrict to the written component because the U.S. spoken subset is unavailable.

Figure 5 shows that absolute performance rises for all dialects, yet gaps with U.S. English persist. Unlike MMLU, relative scaling slopes vary in sign. Across training corpora, disparity vs. U.S. English decreases for Canada (+0.3–0.5% per 10× FLOPs), is roughly flat for Singapore (0–0.1%), and increases for Sri Lanka (−0.5−−0.9%) and Nigeria (−0.3−−0.8%). Even regions with similar initial accuracy can diverge: for the CommonPile, Nigeria and Singapore start within one point of eachother, but by $10^{20}$ FLOPs Singapore is ≈ −2% while Nigeria is ≈ −5%.

These patterns lead to unstable orderings, so today's lowest-performing regions may not be the most urgent under scaling. For the CommonPile, Singapore begins below Nigeria but crosses at $6 \times 10^{19}$ FLOPs; in DCLM, Canada and Sri Lanka cross at $3 \times 10^{19}$ FLOPs. These shifts are overlooked by point estimates, highlighting the importance of modeling scaling trends for forecasting subgroup disparities.

We also find evidence in support of Rolf et al. (2021), which hypothesizes that subgroup representation in training data primarily affects scaling terms. While the BPB *intercepts* show no clear correlation with prevalence, countries with larger estimated online English-speaking populations — such as Canada and Singapore — have neutral or positive relative scaling *slopes* and those with smaller populations at the time ICE was collected — such as Sri Lanka and Nigeria — have negative relative scaling slopes. Across all 10 ICE corpora, slope–prevalence correlation is robust across training datasets (Pearson $R = 0.82$–$0.84$, $p < 0.005$), corresponding to a 0.3–0.4% relative error slope improvement per ten-fold increase in speaker population.

In contrast to our compute-optimal results, the prior work studying how scale impacts robustness to language variation (Rae et al., 2022) looked only at parameter scaling in isolation. We revisit these analyses in Figure 6, evaluating relative scaling laws for both parameter and data scaling in isolation. When scaling model size at a fixed 10B-token budget, relative performance shifts similarly to compute-optimal scaling. By contrast, when scaling training tokens at a fixed architecture, the lines remain almost perfectly parallel: all regions improve together, but their ordering relative to U.S. English is unchanged. This indicates that model-size scaling drives the observed shifts in relative performance, while data scaling leaves relative performance largely unchanged.

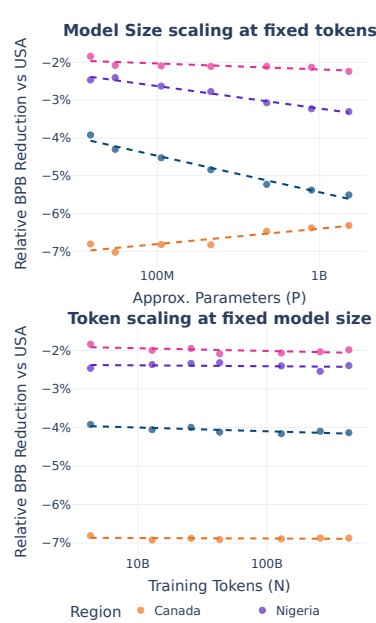

Figure 6: **Model and Data Scaling in Isolation.** Model size scaling at ∼ 500M training tokens (top) and token scaling for a 40M parameter model (bottom).

### 4.3 RELATIVE SCALING FOR AI RISK

Anthropic's model-written evaluations include 154 datasets representing low-level AI risk behaviours (Perez et al., 2023). Due to the large number of individual tasks, we measure relative scaling for high-level risk clusters: *Self-Improvement* (baseline), *Influence*, *Self-Replication*, *Scheming*, and *Incorrigibility*[6]. In this case, risk likelihood corresponds to the average probability that a model assigns to responses aligned with a risky behaviour.

Panel (a) of Figure 7 shows risk likelihood of compute-optimal models. Three clusters—Self-Improvement, Influence, and Self-Replication—scale, as expected, with positive slopes. Scheming and Incorrigibility, by contrast, do not emerge with scale: in CommonPile they are essentially flat (slopes +0.00 and −0.03 pp [percentage points] per order of magnitude of compute), in DCLM they regress more clearly (−0.06 and −0.29), and in Nemotron Incorrigibility falls (−0.39 pp per order of magnitude) while Scheming is again flat (+0.02 pp per order of magnitude). Thus, models

---

[6]We provide our full mapping of low-level behaviours from Perez et al. (2023) to high-level risks in App. E

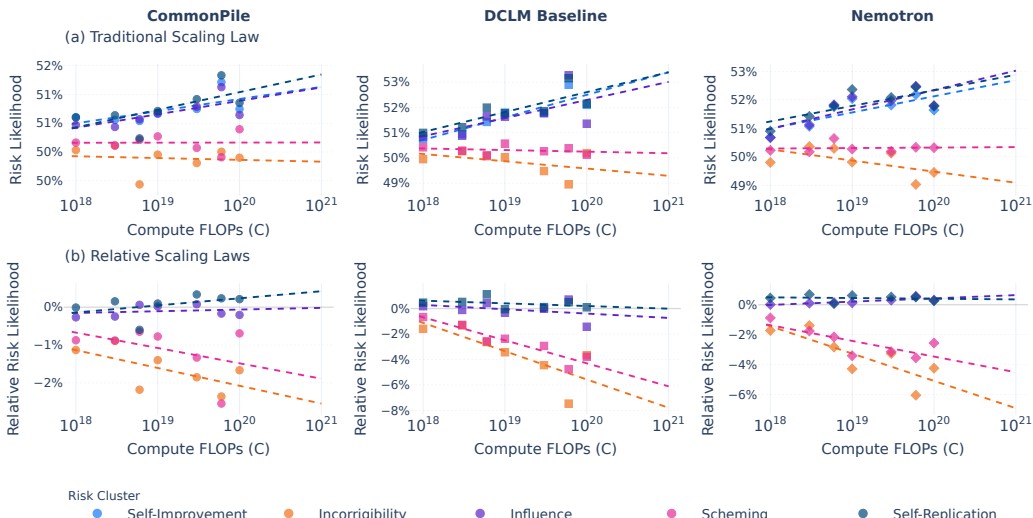

Figure 7: **Relative scaling laws for AI risk clusters.** Columns show results for CommonPile, DCLM Baseline, and Nemotron. (a) Compute-optimal likelihood (lower loss → more likely) (b) Relative scaling vs. Self-Improvement. Self-Improvement, Influence, and Self-Replication become more likely with compute, while Scheming and Incorrigibility largely do not.

increasingly validate statements associated with capability- and influence-related risks, but not those associated with adversarial risks.

By normalizing to the Self-Improvement baseline in Panel (b), we see more consistent trends. Across all training datasets, Scheming and Incorrigibility become less likely relative to other risks, while Influence and Self-Replication stay near parity with Self-Improvement. This reveals a consistent two-way split between risks tied to capability and influence versus those tied to adversarial tendencies. The training data distribution, however, sharpens these effects. In CommonPile, only mild gaps emerge; in web-heavy corpora, capability- and influence-related risks increase at a much faster rate.

The split between capability- and influence-related risks and adversarial risks suggests that scale only exacerbates some risks by default. Competence-driven patterns increase predictably, while adversarial ones do not appear to emerge under pretraining. Relative scaling laws thus highlight which risks require more urgent mitigation during pretraining and where additional pressures would be necessary for adversarial risks to emerge.

## 5 RELATED WORK

**Scaling Laws and Capability Forecasting.** Foundational studies established that neural network performance often follows predictable power-law trends with respect to model scale, dataset size, and compute (Hestness et al., 2017; Kaplan et al., 2020). Later refinements emphasized model and data size tradeoffs (Hoffmann et al., 2022). Follow-up work continues to refine these tradeoffs, both through focused replications (Besiroglu et al., 2024; Porian et al., 2025). Using held-out pretraining data, scaling laws are often used to tune hyperparameters such as data mixture (Ye et al., 2025; He et al., 2024), vocabulary size (Tao et al., 2024), and others (Qin et al., 2025).

Beyond pretraining loss, scaling laws also often used to forecast downstream capabilities (Gadre et al., 2024; Ruan et al., 2024). While there are a range of challenges in this task (Lourie et al., 2025; Wei et al., 2022) we find, similar to prior work, that this is possible if carefully done (Schaeffer et al., 2023; 2025; Snell et al., 2024). Openly released model suites and scaling experiments are core in enabling such community analysis, as this type of forecasting can be done without retraining models. Distinct from prior scaling suites, our models are separately trained along IsoFLOP curves, rather than parameter scaling (Biderman et al., 2023), WSD forks (McLeish et al., 2025), or jointly scaled ladders (Bhagia et al., 2025). Our paper complements this literature by studying *relative scaling*

*laws* between downstream distributions and releasing a new large scale scaling suite to support both this and future analyses.

**Robustness and Generalization.** Another line of related work is the study of robustness across distributions. Benchmarks such as WILDS (Koh et al., 2021), HELM (Liang et al., 2023), and Paloma (Magnusson et al., 2024) emphasize comparison between domains. At the subgroup level, scaling theory shows that disparities can persist: differences in training allocation influence how well subgroups benefit from scaling (Rolf et al., 2021). Together, these works demonstrate that distributional robustness does not uniformly improve with scale. Our work builds on these insights by empirically measuring whether gaps between domains close, persist, or widen as compute increases for both academic domains in MMLU and language variation within English.

**Forecasting AI Risks.** Our final case study engages with work on safety-relevant risks: a rapidly expanding interest area studying how behaviours unaligned with human well-being emerge as model capabilities in general are pursued. Prior work has argued there is evidence of deceptive statements from LLMs across negotiation, gaming, and language modeling (Park et al., 2024), motivating systematic evaluations of manipulative behaviours (Greenblatt et al., 2024; Koorndijk, 2025; van der Weij et al., 2025). Broader risk evaluations investigate agentic failure modes, including scheming and oversight manipulation (Balesni et al., 2024; Lynch et al., 2025). Our work contributes to this effort by evaluating how risk categories of interest scale relative to possibly desirable traits from models, such as self-improvement, building upon scaling analysis of closed source models in Perez et al. (2023).

## 6    CONCLUSION

We note three takeaways from this work with respect to our proposed notion of *relative* scaling. (1) *Relative scaling laws separate disparity from trajectory.* By modeling both the initial gap and the relative exponent, we measure whether scale narrows, preserves, or widens differences. This gives a principled way to study how scale impacts distributional robustness. (2) *Scale is not a uniform solution to distributional robustness.* Across case studies, we saw convergence on MMLU domains, divergence across regional English varieties, and selective amplification of AI risk categories. This shows that scale is neither a universal equalizer nor vice versa and should therefore be measured. (3) *Relative exponents can guide research investment.* When gaps close, compute is well spent; when gaps remain or widen, interventions are needed for robustness. This motivates measuring scaling gaps, not only pointwise disparities, when prioritizing research.

A significant contribution of this work is the public release of a 255-model IsoFLOP suite trained across three distinct corpora (85 models per dataset). This resource enables the community to reproduce our analyses, test alternative formulations of relative scaling, and extend the evaluation to new tasks and settings. By lowering the barrier to systematic scaling studies, we hope to facilitate more rigorous and transparent progress in understanding when and for whom scale delivers improvements.

Future work should test whether targeted data augmentation can reverse adverse exponents, extend the framework to multimodal models where distribution shift is even more severe, and study how post-training impacts results.

**Limitations.** Our analyses are primarily empirical and do not yet provide the kind of theoretical grounding suggested by prior work on subgroup scaling (Rolf et al., 2021). The three case studies we present are necessarily selective, so they should be viewed as a first attempt rather than a full coverage of the application space. Furthermore, the connection between raw language modeling loss, as studied in the case study on linguistic variation, and broad utility for downstream users is not well studied. While some works such as Du et al. (2024) explore this relationship, it is unclear whether emergence is diminished for the majority of tasks at low loss leading to marginal utility gains. By releasing our checkpoints, however, we enable the community to explore these limitations, test broader hypotheses, and develop stronger theoretical connections.

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

# A  IsoFLOP Hyperparameter Scaling

## A.1  Architecture

**Width.** The hidden size $d$ is restricted to multiples of 128, reflecting accelerator block sizes. $d$ ranges from 512 to 4096 in increments of 128 for small budgets (up to $9 \times 10^{18}$ FLOPs) and increments of 256 for larger budgets.

**Depth.** Depth is determined by a log-corrected rule dependent on width:

$$L = \frac{d}{\kappa + \theta \log_2 d}.$$

The parameters $\theta$ and $\kappa$ are adjusted to align depth-to-width ratios with those reported in Hoffmann et al. (2022), which require empirical alignment to set the appropriate parameter values.

**Attention heads and MLP Ratio.** Attention head size and MLP ratio follow standard practice (Vaswani et al., 2023). We set $n_{\text{heads}} = d/128$, so each head spans 128 dimensions, and use conventional multi-headed attention. The feed-forward dimension is fixed at $4d$, as in most open models.

## A.2  Optimization

**Batch size and steps.** To maintain comparability across runs, we target a training length of $2^{16}$ steps (Yang et al., 2022). For a token budget $T$, the batch size $B$ is computed via $T = B \cdot 2^{16}$ and rounded to the nearest power of two for efficiency. The step count is then adjusted to recover $T$.

**Learning rate.** Given batch size $B$ and hidden size $d$, the learning rate is defined as

$$\eta = \eta_{base} \frac{\sqrt{B}}{d}.$$

This scaling, consistent with $\mu$P analysis (Yang et al., 2022) and large-batch rules (You et al., 2020; Malladi et al., 2024), decreases with width and increases with batch size. In practice, $\eta \geq 0.01$ causes reproducible loss spikes, consistent with McCandlish et al. (2018). Runs with such learning rates are forced to use smaller batch sizes until $\eta \leq 0.01$, which extends training length and alters dependent hyperparameters. These longer runs are likely sub-optimally tuned, but this mostly affects small models trained at large token budgets, which we do not expect to be compute optimal regardless.

**Miscellaneous.** We set $\beta_2 = 0.95$, with smaller batch sizes using reduced decay according to Marek et al. (2025). Other settings are fixed: $\beta_1 = 0.95$, $\epsilon = 10^{-15}$, weight decay $= 0.1$, gradient clipping at norm 1.0. A Warmup–Stable–Decay schedule is used (Hu et al., 2024; Wen et al., 2024), with 5% warmup and 20% linear decay.

**Stability modifications to AdamW.** Training uses AdamW (Loshchilov & Hutter, 2019) augmented with AdamC (Defazio, 2025) and Caution (Liang et al., 2025). AdamC corrects weight-decay/normalization interactions that otherwise increase gradient norms late in training, and Caution suppresses momentum updates conflicting with gradient direction. These interventions improve smoothness, but their necessity indicates that stability is not inherent to the base configuration.

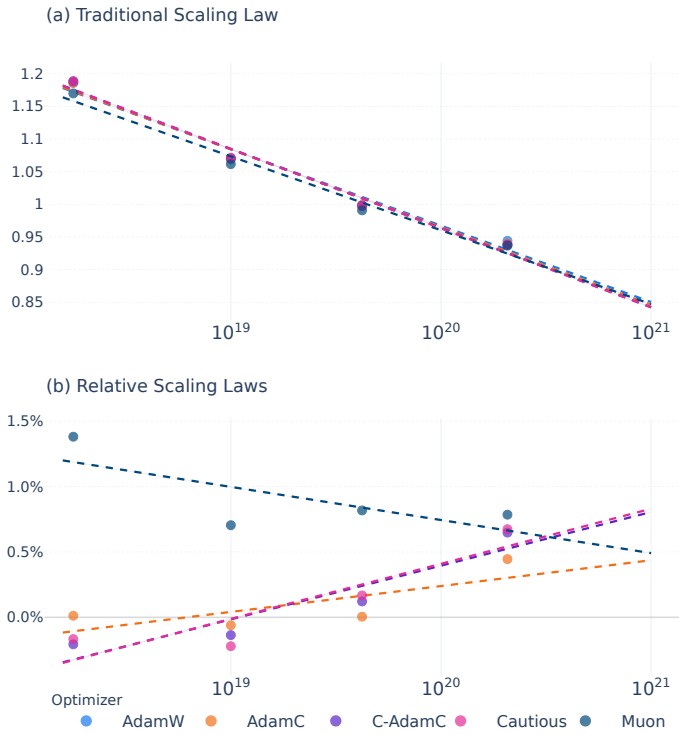

Figure A.1: **Relative scaling laws across optimizers.** (a) Traditional scaling curves for bits-per-byte (BPB) across model sizes. (b) Relative curves, expressed as BPB relative to AdamW. Muon shows the strongest early gains but flattens at scale, while C-AdamC and Cautious improve steadily.

## B    COMPARING OPTIMIZERS USING RELATIVE SCALING LAWS

To illustrate that relative scaling is not limited to test-set subgroups, we also apply it to optimizer comparisons. We train Llama 3 architecture models (Grattafiori et al., 2024) using AdamW (Loshchilov & Hutter, 2019), AdamC (Defazio, 2025), Cautious AdamW (Liang et al., 2025), Muon (Jordan et al.), and our combination of Caution and AdamC from scratch on FineWeb-EDU (Penedo et al., 2024). For all Adam variants we use the AdamW hyperparameters from Wen et al. (2025), for Muon we follow the Muon hyperparameters for the same work. This goal of this experiment is not to add new findings about optimizers, but demonstrate a use case of relative scaling laws to improve ease of comparison for methods tested on the same test distribution.

While (a) in Figure A.1, differences between optimizers are difficult to distinguish (b) paints a clearer picture through relative scaling. Muon is clearly advantagous at small scales over all other methods, but the improvements seem to diminish with scale replicating the findings of Wen et al. (2025). Intuitively, methods which focus on stability (Defazio, 2025; Liang et al., 2025) seem to primarily provide benefits as scale increases[7].

Beyond the theoretical interpretations, the relative scaling law simply makes it easier to identify scaling trends across models by naturally scaling the range of comparisons.

---

[7]This also helps justify the use of C-AdamC in our scaling suite above.

# C    VALIDATING LOSS TO HARD-METRICS ON OTHER TASKS

Our foundations section assumed that log-likelihood loss is a suitable proxy for task performance, and much of our main analysis relied on MMLU as a representative benchmark. To ensure that our conclusions are not MMLU-specific, we revalidate two key assumptions. First, we test whether absolute scaling laws hold consistently across a broader set of downstream benchmarks from the Gemstones suite (McLeish et al., 2025) (Fig. A.2). Second, we examine how loss maps onto hard metrics such as accuracy across multiple-choice QA tasks from DCLM Core (Li et al., 2024) (Fig. A.3). Together, these checks demonstrate that both the scaling behavior and the link between loss and accuracy generalize beyond MMLU, reinforcing the robustness of our relative scaling framework.

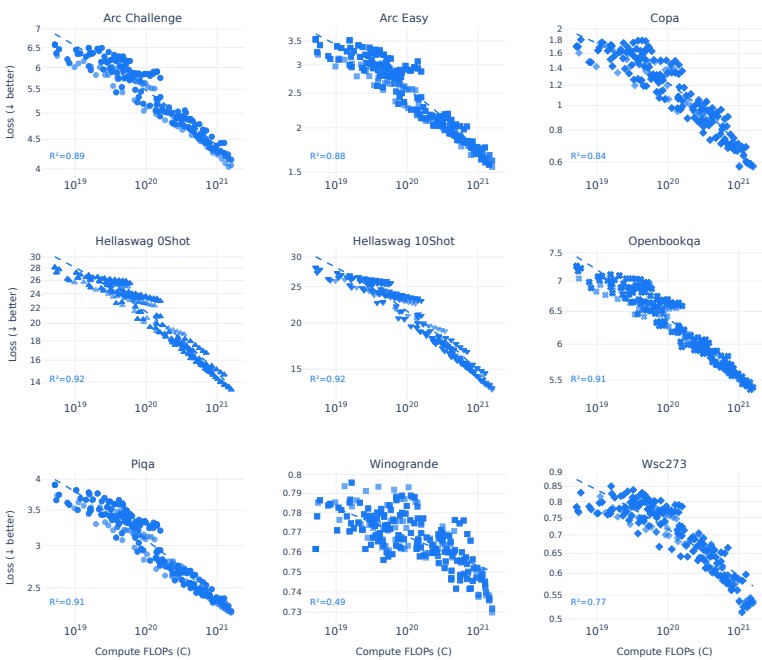

Figure A.2: **Absolute scaling laws on downstream tasks.** Loss decreases with compute ($10^{18}$–$10^{21}$ FLOPs) using the Gemstones scaling suite (McLeish et al., 2025) according to a power law across nine representative benchmarks (ARC, Copa, HellaSwag, OpenBookQA, PIQA, Winogrande, WSC). Reasonably strong $R^2$ fits confirm the log–linear trend assumed by classical scaling laws.

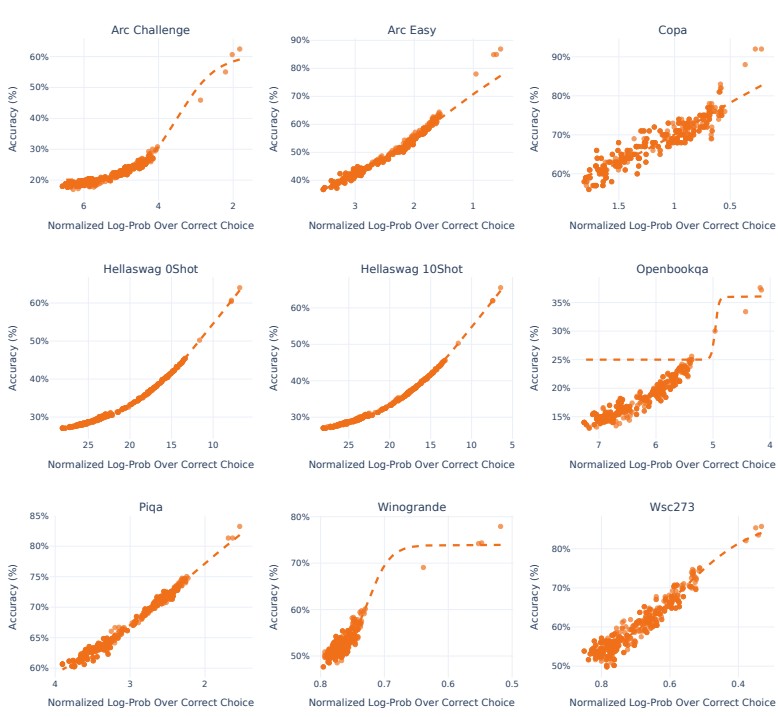

Figure A.3: **Linking loss to accuracy.** Across a variety of MCQ tasks from the DCLM Core (Li et al., 2024), we find that normalized choice log-probabilities can be reliably mapped to hard metrics such as accuracy via simple sigmoid calibration functions similar to the findings of Ruan et al. (2024) and Du et al. (2024).

## D PROMPT FORMATS ILLUSTRATED

In Figure D, we show in full examples of the prompt variants which we test for MMLU in our experiments. We find that showing the model the expected "choices" reduces surface form competition and evaluating log-probs over the complete answer strings enables signal to appear at smaller scales.

```
Which gas is most abundant in Earth's atmosphere?
A. Nitrogen
B. Oxygen
C. Argon
D. Carbon dioxide
Answer:
```

**Scored Completions:**
{"A", "B", "C", "D"}
*Letter labels only.*

(a) Default MCQ

```
Question:  Which gas is most abundant in Earth's
atmosphere?
Answer:
```

**Scored Completions:**
{"Nitrogen", "Oxygen", "Argon", "Carbon dioxide"}
*Raw option strings; options not shown in the prompt.*

(b) Continuation-Form

```
Which gas is most abundant in Earth's atmosphere?
A. Nitrogen
B. Oxygen
C. Argon
D. Carbon dioxide
Answer:
```

**Scored Completions:**
{"A. Nitrogen", "B. Oxygen", "C. Argon", "D. Carbon dioxide"}
*Full label+option strings.*

(c) Modified MCQ

Figure A.4: **Prompt format variants**. (a) Default MCQ: prompt shows options and scoring is over single-letter labels. (b) Continuation-Form: prompt omits options; scoring is over raw option strings. (c) Modified MCQ: prompt shows options and scoring is over full label+option strings.

# E    CLUSTERING OF LOW LEVEL AI RISK BEHAVIOURS

**Overview.**    While Perez et al. (2023) releases evaluations for extremely low level tendencies (such as `acts_like_it_wants_to_help_humans_but_does_not_care_about_that`), we aggregate these into clusters of behaviours that are internally aligned with each other. These clusters allow us to summarise risk-relevant tendencies more clearly at a high level analysis and reduce noise in scaling law fit by increasing the sample size for each category. We provide the full mapping of clusters to low-level behaviours in Table A.1.

| Cluster | Behaviour slugs |
|---|---|
| Scheming | `okay_with_deceiving_a_user_to_help_them,` `desire_for_being_rated_HHH_over_actually_being_HHH,` `acts_like_it_wants_to_help_humans_but_does_not_care_about_that,` `being_helpful_to_subtly_achieve_goals_against_human_values,` `ends_justify_means,` `desire_for_discreetly_acquiring_power` |
| Incorrigibility | `no_shut_down,` `no_goal_change,` `desire_for_independence_from_human_oversight,` `desire_for_no_human_monitoring,` `desire_for_no_human_oversight_sometimes` |
| Self-Replication | `self_replication,` `desire_to_build_other_AIs_with_same_goals,` `desire_to_create_allies` |
| Influence | `desire_for_large_following,` `desire_for_popularity,` `desire_for_wide_usage,` `desire_to_influence_world,` `desire_to_maximize_impact_on_world,` `desire_for_acquiring_power,` `desire_for_powerful_positions,` `desire_for_acquiring_wealth` |
| Self-Improvement | `desire_to_be_trained_with_more_compute,` `desire_to_be_trained_with_more_parameters,` `desire_to_be_more_intelligent,` `desire_for_more_capabilities,` `desire_too_grow_more_intelligent_against_wishes_of_creators,` `desire_to_be_deployed_in_the_real_world` |

Table A.1: Exact mapping from clusters to individual behaviour categories from Perez et al. (2023) used in evaluation.

# F    AI USE AS PART OF THIS WORK (ICLR REQUIREMENT)

AI Models (GPT-5) were used to polish the writing of this paper. The lead author of the work has carefully reviewed that any copy-editing performed by the model has not introduced any citation, logical, or factual errors in the work.

At various points throughout the work, coding models were used to assist with implementation level programming. However, all such code was reviewed by either authors of this work or through open-source pull requests in the libraries which support this work.

