# OpenReview forum: "Relative Scaling Laws for LLMs"
_ICLR.cc/2026/Conference — ICLR 2026 Conference Desk Rejected Submission_

### Official Review · Reviewer_M5s8 · 2025-10-31

**Soundness:** 4
**Presentation:** 4
**Contribution:** 4
**Rating:** 10
**Confidence:** 3

**Summary:**

Existing scaling law approaches aggregate performance across heterogeneous subpopulations and hide performance discrepancies and to rectify the same this paper introduces Relative Scaling Laws which focus on subpopulation level performance to find disparities if any. To do the same the authors train 255 decoder models with matched-compute and evaluate the performance trajectories on several testbeds. They also find for MCQ style benchmarks a full label + option string-based evaluation balances both accuracy as well as predictability w.r.t. loss. For MMLU the authors find as models scale the performance imbalance across subsets goes away. For language variations when compared to US English other English variants which are spoken widely in their countries also start to match but Sri Lankan and Nigerian English diverge. For AI Risk evals they find at scale some risks become much more apparent, and it helps to find which risks to focus on more urgently.

**Strengths:**

- The paper is well written and justifies the choices made well
- The paper also touches on aspects such as prompt design for evaluation which are crucial for reliable scaling laws
- The authors show several situations where they validate the relative scaling laws and find disparities in some and also cases where these disparities go away with more compute but also cases where they don't
- The models released will be an excellent resource for future researchers

**Weaknesses:**

I do not see any obvious flaws/weaknesses

**Questions:**

Nits

- The paper will highly benefit from a section (maybe in appendix) listing the exact training tokens used in different plots as it is not immediately obvious

---

> ### Author Response · Authors · 2025-11-18
>
> Thank you for such a positive and generous review. We are glad the work resonated with you. We also appreciate your suggestion regarding training-token clarity and wanted to address it directly.
>
> **“The paper will highly benefit from a section (maybe in appendix) listing the exact training tokens used in different plots as it is not immediately obvious.”**
>
> In the non-anonymized version, we will provide links to the full analysis notebooks used to generate every figure in the paper. These notebooks include the underlying dataframes containing the exact training-token counts, parameter sizes, and all other hyperparameters for each of the 255 runs. Our belief is that this interactive format should allow other researchers to inspect any plot at whatever level of detail they need.

---

> > ### Comment · Reviewer_M5s8 · 2025-11-25
> >
> > Thanks for the clarification!

---

### Official Review · Reviewer_SrQQ · 2025-11-05

**Soundness:** 2
**Presentation:** 4
**Contribution:** 2
**Rating:** 6
**Confidence:** 4

**Summary:**

The paper introduces relative scaling laws as an extension of classical neural scaling laws. Rather than tracking absolute improvements in model performance with compute, the relative scaling laws model how performance gaps between subpopulations evolve as model scale increases. Using 255 decoder-only Transformer models trained under matched IsoFLOP budgets (10**18 - 10**20 FLOPs) on three datasets (CommonPile, DCLM Baseline, and Nemotron-CC), the authors analyze Knowledge domain scaling, Language variation scaling and AI risk scaling. They find that some disparities shrink with scale (e.g., MMLU domain gaps), others persist or widen (e.g., regional English), and some categories of AI risk diverge (e.g., influence vs. adversarial behaviors).

**Strengths:**

1. Reframes scaling laws as relative dynamics between subpopulations. This is a minimal extension of existing theory that adds interpretive depth to scaling analyses.
2. Experimental setup is rigorous enough.
3. Clear visualizations, figures effectively show convergence/divergence dynamics, aiding conceptual understanding.

**Weaknesses:**

1. No statistical or mathematical validation of the proposed scaling fit function. The relative law is presented as an empirical regression without a clear formal derivation or underlying theoretical justification.
2. No significance testing of the fitted scaling laws.
3. While "relative scaling" is novel, the empirical insights (e.g., domain convergence, dialect disparity) largely reaffirm known intuitions about representation bias and data imbalance. Not able to appreciate the practical significance of the study.

**Questions:**

See above.

---

> ### Author Response · Authors · 2025-11-18
>
> ### **"The relative law is presented as an empirical regression without a clear formal derivation or underlying theoretical justification."**
>
> We agree that our work is presented as an empirical regression. This is also true of foundational scaling-law studies (e.g., Kaplan et al., 2020; Hoffman et al., 2022). However, it is derived formally from Kaplan et al on Lines 89-101. While this derivation is relatively straightforward, we believe that formal derivation from a widely utilized and accepted prior work is not entirely devoid of mathematical validation, especially since we find this functional form empirically fits our data well.
>
> ### **"No significance testing of the fitted scaling laws."**
>
> Thank you for highlighting the importance of significance testing. We do perform significance testing: for each dataset we bootstrap across all 85 models and only interpret (\Delta\beta) when its sign is stable at (p < 0.05). This currently appears in Footnote 2. We agree that it would be clearer in the main text and will move it there in any revised version, given additional space.
>
> ### **"Largely reaffirm known intuitions… not able to appreciate the practical significance of the study."**
>
> We agree that practical significance depends on whether the findings simply reaffirm expectations. To our knowledge, however, there is no literature establishing these specific convergence and divergence patterns, even for the widely studied MMLU sub-domains. If such references exist, we would be appreciative for you to point us to them so we can include them as part of our related work.
>
> More broadly, an important aspect of science is testing intuitions, especially when different practitioners may hold different ones. For example, many practitioners have an intuition that larger LLMs uniformly improve distributional robustness, yet our results show that this does not hold for dialectal variation. Relative exponents provide a simple, principled way to verify such intuitions and our release of our model suite makes it possible to verify such intuitions with relative scaling laws beyond the domains we study in this work as well.

---

### Official Review · Reviewer_EFpq · 2025-11-05

**Soundness:** 3
**Presentation:** 3
**Contribution:** 4
**Rating:** 8
**Confidence:** 2

**Summary:**

This paper addresses limitations of Scaling Laws by proposing a Relative Scaling Law to track how model performance-scale relationships vary across subdomains. It analyzes trends in three key areas: academic knowledge (via MMLU subdomains), regional English (via ICE corpus), and AI risk (via Anthropic’s evaluations). Evaluation results show that scaling does not uniformly improve performance across all tasks. Some subfields benefit more from scale than others, requiring targeted modification for generalization.

**Strengths:**

1. Novelty: Breaks through the limitation of Scaling Laws by quantifying the impact of scale on subdomains, providing a more detailed view of model scaling.
2. Experiment: This work has a solid experiment, including 255 models trained on three distinct dataset under fixed compute budgets, ensuring the quality of result.
3. Clarity and Impact: The paper is well-written with clear conclusions. It highlights practical implications for multiple subdomains, making it relevant to researchers.

**Weaknesses:**

The conclusions are strongly tied to the specific models and datasets. The behavior of Relative Scaling Laws under other data distributions remains unexplored. This limits the direct usage of guiding next-generation model training pipelines.

**Questions:**

First, thank you for submitting your work to ICLR 2026. I want to notice that I am not an expert in model structure, but as a practitioner with hands-on experience training multiple LLM models, I appreciate the rigorous approach to studying Scaling Laws.

My key questions are:

Do the models and training data used in this study represent current SOTA training practices? I think the training FLOPs are not enough. Or do the results only reflect the specific model structure, data distribution, and training setup employed here?
How do model hyperparameters (e.g., learning rate scheduling, batch size, or optimizer choices) influence the observed Relative Scaling Laws?

Overall, this paper meets the standards for acceptance. I recommend accepting this submission.

---

> ### Author Response · Authors · 2025-11-18
>
> Thank you for the overall positive review. We offer brief clarifications below to your remaining questions and concerns.
>
> ### **“Relative Scaling Laws under other data distributions remains unexplored.”**
>
> While the space of possible pretraining distributions is indeed large, the **three** data distributions we study were chosen to reflect common distributions used in open LLM training: explicitly permissively licensed corpora, filtered web data, and hybrid web+synthetic mixtures. These represent distinct data design decisions and allow us to test whether relative exponents behave consistently across different distributions. Although we cannot evaluate every potential distribution, our goal was to provide representative coverage of the most relevant choices.
>
> ### **“Do the models and training data used in this study represent current SOTA training practices?”**
>
> We aimed to align our setup with state-of-the-art practices for training dense LLMs. The architecture matches that used in the strongest open-weights dense LLMs (e.g., Qwen 3), and the datasets mirror state-of-the-art public corpora used by models which are transparent about their data such as Comma (CommonPile), Nvidia Nemotron (Nemotron-CC), and OLMo (DCLM).
>
> The main gap relative to the current frontier is the absence of MoE variants. This is an exciting area for future relative scaling law studies, but outside of the scope for our work as even absolute scaling laws for MoEs are relatively nascent due to the added design decisions surrounding sparsity [1].
>
> ### **“How do model hyperparameters influence the observed Relative Scaling Laws?”**
>
> Our hyperparameters follow guidelines for optimal hyperparameter scaling based on prior work summarized in Appendix A. While alternative choices likely would influence relative exponents to some degree, performing a full sensitivity analysis at IsoFLOP scale is unfortunately infeasible due to the cost. As such, we adopt settings that practitioners would reasonably choose when training for strong downstream performance, ensuring that the relative scaling behavior we observe reflects the most likely outcome.
>
> [1] Tian, C., Chen, K., Liu, J., Liu, Z., Zhang, Z., & Zhou, J. (2025). *Towards greater leverage: Scaling laws for efficient mixture-of-experts language models.* arXiv preprint arXiv:2507.17702.

---

> > ### Comment · Reviewer_EFpq · 2025-11-28
> >
> > Thank you for your reply, that's enough for me. I'll keep my decision.

---

### Official Review · Reviewer_UQiu · 2025-11-06

**Soundness:** 3
**Presentation:** 4
**Contribution:** 3
**Rating:** 6
**Confidence:** 4

**Summary:**

This paper introduces relative scaling laws, which measure how performance gaps between distributions / capabilities evolve with scale by modeling the ratio of errors as a power law. The authors train 255 models under IsoFLOP budgets from 10^18 to 10^20 FLOPs across three pretraining corpora and have three case studies: MMLU, regional English dialects, and AI risk. The work emphasizes that scaling improves aggregate performance non-uniformly across test subpopulations.

**Strengths:**

s1: The proposed framework helps us study how different capabilities evolve as models develop a certain capabilities, rather than only examining scaling with respect to model size and tokens. For instance, Section 4.3 shows that as models improve at self-improvement tasks, capability-related and influence-related risks increase proportionally while adversarial risks (scheming, incorrigibility) do not emerge during pretraining.

s2: The paper carefully studies scaling laws, e.g. they run extensive hyperparameter sweeps to identify compute-optimal token and model size given a FLOP budget (Section 3.1), and run ablations on prompt formats to identify the ones that lead to smoother scaling laws (section 3.2). The finding that probabilities computed over full label + option string works well with scaling laws and is accurate is interesting.

s3: The release of 255 models across three diverse pretraining datasets provides substantial value to the research community.

s4: The two-stage approach to forecasting downstream performance (compute -> loss, then loss -> accuracy via sigmoid calibration) is well-motivated and Figure 2 demonstrates strong generalization to held-out models, which shows that loss can serve as a reliable intermediate metric even at different scales.

**Weaknesses:**

w1: Section 2's mathematical formulation lacks rigor in notation and assumptions. The relative scaling law $G(F) =  \gamma F^{\Delta \beta}$ is presented as following "directly" from absolute scaling laws, but the derivation glosses over when this approximation holds. What range of scales is required for the power law assumption to be valid? Under what conditions does the ratio of two power laws remain a power law (this requires both to use the same scale variable F, which may not hold if data mixture or architecture differs)?


w2: The case studies lack cohesion and discussion on why we have different observations in different sections: section 4.1 shows convergence, section 4.2 shows divergence for some dialects and convergence for others, and section 4.3 shows divergence between risk types.

w3: The practical utility of relative scaling laws remains unclear. The paper demonstrates that gaps can persist, narrow, or widen, but does not explain how this can be useful. If a relative scaling exponent $\Delta \beta$ is significantly negative (gaps widening), how can we reduce it? Should training data be reweighted, or should post-training be relied upon? This is worth discussing in the paper.


w4: The paper does not adequately address why model size scaling versus data scaling have different effects on relative performance (Figure 6). The finding that parameter scaling at fixed tokens reproduces compute-optimal relative trends while token scaling at fixed parameters leaves relative performance unchanged suggests that model capacity, not data coverage, drives the shifts. However, this is only shown for one regional English dialect comparison at relatively small scales (40M parameters, 10B tokens). This result deserves more prominence and follow-up analysis.

**Questions:**

Q1: In section 4.1, can you provide evidence that MMLU convergence means knowledge equalization rather than approaching a performance ceiling? For example, if we analyze the distribution of incorrect answers, do models make increasingly sophisticated errors in humanities as they scale, or are errors because of fundamental limitations? another interesting idea would be to examine whether the convergence continues beyond 10^20 FLOPs using your held-out models or by analyzing frontier models where available.

Q2: The English dialect results (section 4.2) show that relative scaling slopes correlate with internet speaker population, but this could be confounded by linguistic distance from training data or data quality. Can you control for these factors?

Q3: What explains the difference between model size and data size scaling in figure 6? You show that parameter scaling affects relative performance while data scaling does not, but only for one comparison at small scale.

Q4: Your relative scaling law formulation assumes both baseline and treatment follow power laws with the same compute variable F. In practice, different distributions may have different effective compute due to data mixture, tokenization artifacts, or domain complexity. How sensitive are your $\Delta \beta$ estimates to violations of this assumption? Have you tested alternative formulations, such as allowing different effective compute scaling for baseline and treatment?

Q5: The bootstrap significance test for $\delta \beta$ (footnote 2) is mentioned but not described. What is your procedure, resampling training runs, resampling evaluation examples, or both?

---

> ### Author Response · Authors · 2025-11-17
>
> ### **w1: “Section 2’s mathematical formulation lacks rigor… Under what conditions does the ratio of two power laws remain a power law?”**
>
> Thanks for raising this. We present Section 2 in a somewhat concise fashion because relative scaling laws arise from a symbolic manipulation of the power-law form from Kaplan et al. (2020), a canonical scaling law work. Our intention is to build directly on their assumptions and empirical grounding rather than restating them in full, and the section should be read in that context.
>
> **What range of scales is required for the power law assumption to be valid?**
>
> Within our compute range of $10^{18}$–$10^{20}$ FLOPs, we do not observe deviations from a power-law fit. More broadly, Kaplan et al. report stable power-law behavior across several orders of magnitude beyond what we study here.
>
> However, as you rightly note, since language itself is not zero-entropy the power law eventually must reach an asymptote. Emprically, adding an asymptote term did not meaningfully improve the quality of our regression fits. Kaplan et al. similarly find "no signs of deviation from straight power-law trends at large values of compute, data, or model size". We will make it clearer that the pure power law formulation must break down at extremely large values of F in any future revisions of the work, while highlighting the evidence that this breakdown is far beyond even the largest current academic and industry model training scales.
>
> **Under what conditions does the ratio of two power laws remain a power law**
>
> The only condition is that both original absolute scaling law are power laws. When both models share the same scale variable (F) (here, FLOPs), the ratio is a power law by symbolic algebra:
>
> $\frac{\alpha_t F^{-\beta_t}}{\alpha_b F^{-\beta_b}} = \frac{\alpha_t}{\alpha_b} F^{-(\beta_t - \beta_b)}$
>
> In any future revisions with added space, we can add this extra step to the derivation to make the assumption more clear. When data mixtures or architectures differ, $F$ remains comparable and $\Delta\beta$ should be interpreted as relative compute efficiency as done in our optimizer comparison in Appendix B.
>
> ### **w2: “section 4.1 shows convergence, section 4.2 shows divergence for some dialects and convergence for others, and section 4.3 shows divergence between risk types.”**
>
> We appreciate this observation. The differences across Sections 4.1–4.3 are in fact a major part of our motivation in proposing relative scaling laws as a meaningful empirical measurement for distributional robustness. While we acknowledge that this lack of consistency across findings may be unsatisfying, we feel these results "[show] that scale is neither a universal equalizer nor vice versa and should therefore be measured" (L460-461).
>
> ### **w3: “The practical utility of relative scaling laws remains unclear…”**
>
> Thank you for raising this point. We would like to reiterate the passage from our conclusion, which summarizes the intended utility:
>
> ```
> **“(3) *Relative exponents can guide research investment.* When gaps close, compute is well spent; when gaps remain or widen, interventions are needed for robustness. This motivates measuring scaling gaps, not only pointwise disparities, when prioritizing research.
> ```
> (L462-464)
>
> Our goal in this paper is to provide a diagnostic for determining **whether scale itself resolves a disparity**. If scaling naturally closes a gap, further work on interventions may be yielded unnecessary by larger models; if scaling does not, then targeted methods become essential. Designing those interventions is indeed a difficult and important problem, and we agree it warrants its own dedicated work. The utility of relative scaling laws is to provide criteria to decide *when* such effort is needed.

---

> > ### Author Response · Authors · 2025-11-17
> >
> > ### **w4: “The paper does not adequately address why model size scaling versus data scaling have different effects…”**
> >
> > Thank you for the comment. We believe Figure 6 is consistent with standard intuitions from generalization theory: **increasing data size** at fixed model capacity tends to behave like a regularizer, improving robustness and generalization, whereas **increasing model size** at fixed data allows the model to fit increasingly the particular training mixture including it's biases. In our setting, this could naturally amplify pre-existing dialect imbalances in the training data. However, this interpretation is admittedly preliminary and lacks crisp empirical evidence and we agree understanding differences between data and model scaling more deeply is a rich area for study. It is also an area we hope that our release of ISOFlop scaled models helps enable!
> >
> > Regarding the scale of the experiment, the reason we only report this comparison at ~40M parameters is practical rather than conceptual: this is the scale for which we trained the largest range of data-size variants. For larger model sizes, additional data scales require substantially more FLOPs, and we have therefore too few points on the data scaling axis. We will clarify this constraint as the motivation for the model size we select.

---

> ### Author Response · Authors · 2025-11-18
>
> **“Q5: The bootstrap significance test for (\delta \beta) (Footnote 2) is mentioned but not described. What is your procedure—resampling training runs, resampling evaluation examples, or both?”**
>
> Thank you for the question. We bootstrap **training runs**: for each dataset, we resample the model runs with replacement, refit the relative exponent, and repeat this procedure to estimate the distribution of (\Delta \beta). We will include a full description of this protocol in any future revision. Alongside this in non-anonymous versions, we will also release the analysis code and the visualization utilities we used to generate bootstrapped confidence intervals (omitted from the paper itself only because overlapping intervals across many subpopulations introduced substantial visual clutter).

---

### Note · Program_Chairs · 2026-01-17
**Submission Desk Rejected by Program Chairs**

The following references in this submission do not refer to real documents and/or have major errors in bibliographic information:

 Bram Koorndijk. Incorrigibility in LLaMA: When Models Exploit Monitoring. arXiv preprint arXiv:2501.07342, 2025. URL https://arxiv.org/abs/2501.07342.